# Learning Performance Assessment for Culture Environment Learning and Custom Experience with an AR Navigation System

**Yuh-Shihng Chang [1,2,*]** , **Yu-Jhang Rick Hu [3,4] and Hong-Wei Chen [3]**

1   Department of Information Management, National Chin-Yi University of Technology, Taichung 411, Taiwan
2   Department of Inter Design, Asia University, Taichung 413, Taiwan
3   Department of Digital Media Design, Asia University, Taichung 413, Taiwan
4   Department of Visual Communication Design, Chaoyang University of Technology, Taichung 413, Taiwan
*   Correspondence: eric_chang@ncut.edu.tw

**Abstract:** Culturally sustainable environmental education is a crucial issue that is inseparable from our society and environment; and closely linked to our life. Taiwan's cultural environment and family activities are characterized by religious beliefs and temple architectures. Incorporating digital technology into learning about the cultural environment of Taiwan can be an effective way to achieve sustainability. To have a deeper understanding of Taiwan's culture and aesthetics, this study uses AR (augmented reality) technology to bring interactive experiences to the temple, so that the public can interactively experience arts in the temple. Through the AR navigation system, people can ascertain cultural connotations and understand the power of Taiwan's beliefs. This study will take the Wumiao (martial temple) in Yanshui District, Tainan, Taiwan, as the research field to examine the learning experiences and performance with an AR navigation system. The empirical method is used to collect data of users' behaviors and feelings; the statistics approach is also used to testify to the AR usability that is if with AR technology people can have a better understanding and learning performance on the culture aspect. Research finding proves one can have a better learning experience with AR tech with a positive result. According to the correlation coefficient analysis, only perceived enjoyment and perceived ease of use are moderately correlated, the other differences are highly correlated. It is thus clear that AR navigation systems should be applied for having better folk beliefs, cultural etiquette learning experiences, and learning performance, more, for not merely having faith, but also loyalty as well.

**Keywords:** augmented reality; culture environment sustainable education; culture experience

## 1. Introduction

Temples in Taiwan Taiwanese's represent most spiritual commitments, also are the places of worship in which people with the same religion pray together is the places of worship, in which people pray with same religions. Not just for worship, people also celebrate festivals there. Having been important in Taiwan culture always, temples in Taiwan have become the media for local cultures and customs. With advanced digital technologies and popularized media, in Taiwan, many temples have incorporated digital technology to make their own websites, social media, and even application, for promoting their local cultures and customs for people to worship, and for attracting people to take part in their activities. Some temples even take an augmented reality (AR) interactive system to impress people. "Cultural experience (learning)" is one of the socialized development behaviors for people; through experiencing local cultures to convey its connotations and features, spontaneously having

enthusiasm, and start to pay attention in cultural activities. In recent years, Taiwan cultural festivals have become famous tour spots worldwide, for which many temples tend to further promote their own cultures and customs as well. Through AR technology will, they impress people by incorporating interactive digital devices with sacred objects as to help people have better understanding of Taiwan cultures, customs, worship rules, and sacred objects, to provide a better worship experience for prayers.

"Cultural experience" is one of the social behaviors for people to develop and learn; through the way of experience to convey the local cultures, by which one starts to be attentive to cultural activities, with spontaneous enthusiasm that has been cultivated. Culture environment education is focused on learning and practicing the culture-related knowledge, connotation, and lifestyle in a sustainable way [1–3]. Culturally sustainable environmental education may be a contemporary crucial issue since that is inseparable from our society and environment; meantime that is closely linked to our economy and the spiritual level of people living in the real world [4,5].

The cultural experiences at Taiwan's temples expose us to cultural diversity; this increasing cultural diversity reflects Taiwan history and traditions as well as the change in society. Through learning the cultural environment of Taiwanese temples, one will perceive the beauty and connotation in Taiwan's culture. Digital technology has been widely used in education and has achieved good performance. Incorporating digital technology into cultural environment learning can be effectively applied in Taiwanese cultural education to achieve sustainability.

In recent years, digital technology has been applied largely in different fields of education. Especially the education of culture (scene) guide, in which one can see AR, virtual reality (VR) [6], 3D drama, sensor technology, Ellington, and so forth.

This research designed an application for Tainan City Yanshui District martial temple, for people to have an innovative experience; by taking AR technology to provide an interactive experience for people to have a better understanding of the temple's culture and historical background, worship rules, sacred objects, and customs.

The purpose of this research is to collect data of users' behaviors and feelings and then use statistical analysis to prove the usability of the AR technology deployed in Taninan Yanshui District martial temple to promote various cultural experiences.

## 2. Literature Review

### 2.1. Cultures and Religions

People's belief refers to the worship of gods in particular environments such as in temples; this is a concept of co-socialization, spiritual commitment to gods, and expressing hopes that wishes can come true. Following the rules of gods, folk belief in Taiwan results in a diversity of folklore, rituals, beliefs, taboos that form a wide variety of religious cultures and social customs including belief connotations, thoughts, behaviors, and educations. The main religions in Taiwan are Buddhism, Taoism, Confucianism, and so on. People also have worshipped rituals for ancestors, spirits in nature, seasons and agriculture holiday. People worship various gods mainly temples.

Culture and religion are closely linked. In the aesthetics and ethics aspect, religion may identify culture. Accordingly, culture can be recognized by religion [7]. Mulder (1985) stated that religion is a presentation of culture by understanding religion to recognize culture [8]. In Taiwan people's religious community, each god is protective of people from different aspects. For example, Mazu is the Goddess of sea in Taiwan, who likes mother protects people from disasters. There are many tutelary deities who are also protectors or guardians of particular people, things, area, culture, and so forth. Like Hakka Three Lords of the Mountains, and Fujian Bosheng Dadi (Safe Protection Emperor).

This study takes the Taiwan's Tainan Yanshui martial temple as a research guide. Martial temple also called Daguan Dimiao, the worshipped God Guan-Yu, also called Guan-Gong. Martial temple is not merely a famous cultural monument for tour spots but also an original place for peoples festival "Yanshui" firework [9]. For Tainan people, martial temple is a sacred place for worshiping gods.

Having been selected by the Australia's Get Lost magazine as one of the "worldwide ten best festivals", also labeled as the third out of "worldwide ten most dangerous festivals" by Australia's ABC TV, Yanshui firework festival in Taiwan has been popular worldwide for its religious culture and customs. This research thus suggests that incorporating digital technology into martial temple Yanshui firework festivals [9] is a great subject to study. It should provide better interactive experiences for people to have a better understanding of Taiwan's culture (and belief connotations as well).

*2.2. Augmented Reality*

In 1997 Azuma the scholar who was the first person proposing the concept of systematization which lead to today's theory of augmented reality. Nowadays AR has been widely applied and developed by researchers worldwide; the incorporation of AR technology into digital media and mobile devices and diverse businesses have been successfully presented, in the fields of education, entertainment, medicine, military and so forth. One example is the advanced digital technology of the museum guide for conveying information to people in the museum. AR refers to a concept of combining virtual worlds into reality; through a virtual media bringing information to the real world by reinforcing the subjects, or the objects [10,11]. After virtual reality (VR), AR cannot merely bring 3D images to the objects but also get people to watch and feel 3D images from any angle. By using semi-transparent glasses and headset the virtual world and real world are highly connected in which one can see 3D images in real world that makes the interactive experiences more enjoyable.

Taking the field of education as an example it is obvious that after 2D (linear) tech, and the combination of AR and mobile devices makes teaching and learning more creative such as real time algorithms, 3D show, voice and video interactions, message sharing, tagging images, intelligent agent, real time training and so on.

For enhancing AR technology, through 3D glasses headset, Mobile AR has been developed to bring digital into reality [12]. AR technology helps students learn effectively as well as brings unique cultural environment interaction to people [13]. Many scholars have already developed related AR tech for education [14,15]. Besides the part of AR and mobile devices, personal experiences are also important. Teaching design could be seen as a communication; designers will have to understand users' learning environment and lead a conversation [16]. As a matter of fact, users' experiences and content design are the key points for innovative products [17,18]. The most important issue for a communication is to understand users' reaction to their environment. For this issue, some researchers have developed related quantitative sheets which have been used for on line training, e-businesses, and user behavior analysis as well [19]. There are many kinds of environmental quantitative worksheets, different types of interest-triggering content [20] and self-assessment models (SAM) [21]. The efficiency of the quantitative sheets adverted to above has been already proved in research fields [22–24], yet there is not one only for cultural environment learning design and learning performance assessment. For understanding users' reaction to AR in cultural environment learning, this research will examine users' experiences that relate to pleasing values and learning performances; based on Chang et.al.'s research model of (playing) learning performance assessment [14], this research will go further to develop the model and quantitative worksheet.

In addition to the architectures, the core value of martial temple is cultural customs and religious activities. According to field studies, the sacred objects and materials that have been incorporated into AR application including: (1) martial temple God-Guansheng Di Jun statue, (2) dragon door gods, stone lions, wall carvings, traditional objects pillow stones, column bases, and whole building structures, (3) incense burners, worship rules and order, (4) inscriptions and its history, (5) stone pillars and the poems on them, (6) religious activities—Yanshui firework festival. The users can instantly get information by censoring the objects through application, by which cultural learning performance should be enhanced.

*2.3. The Usability Assessments*

"Usability" decides the function for user to experience; usability could be said as a way of communication, by mean of interactive behavior to convey the message to user to recognize and get feedback. The purpose of usability is to improve the users experience and help user communicate with interactive system in life or at work [25]. Thus, the usability of the system provides the user a good operating interface. Kristof and Satran (1995) point out that [26]: (1) the ease of use design has to meet the usability; (2) the functionality: the easier to use, the better. For the digital interaction design study, the information system usability assessment, Hsu and Lin proposed "technology acceptance factors" could be the method to evaluate the users' acceptance degree [27]. Technology acceptance factors include "perceived usefulness", "perceived ease-of-use", and "perceived enjoyment". "Usability" could be defined as a way to decide if the interactive system is useful. "Perceived ease-of-use" could be defined as a way to evaluate how difficult or friendly the interactive system is while the user is using it.

The extended technology acceptance model (TAM2) is proposed by Venkatesh and Davis (2000), they believe that "social impact processes (subjective norms, voluntary, image) affect users' acceptance of technology systems, so "social norms" is included in the extended TAM model [28]. This study also mentioned that religion affects social culture, social literary life and habits, and present people's idea and behavior. Therefore, this study suggests "perceived cohesion" is one of the impact factors of usability. Therefore, the usability assessment is based on the TAM model, regarding the usability of digital technology; "social norms" and "perceived cohesion" related to religion cultural are two factors in the research model. Here are six factors that may decide the usability: (1) perceived enjoyment; (2) perceived cohesion; (3) perceived ease of use; (4) social norms; (5) user preference; (6) user loyalty.

## 3. AR Guide to Application Implementation

Many temples in Taiwan have their own websites and social media for promoting religious culture and customs to people in a digital way. This research takes AR technology to develop guided systems for people to have an interactive experience, just bring about smart phones or mobile devices through this research application, people can have a better understanding of martial temple culture and customs. The software or resources used in this research creation include the following:

(1) Unity 2018.1.5f1: the main developed software of this AR application, it includes the built-in Vuforia Augmented Reality Software Development Kit is the main tool.
(2) Android SDK: to export APK Required Kit of Unity.
(3) Java SE Development Kit 8: to export APK Required Kit of Unity.
(4) Photoshop: the interface software for content creation.
(5) Illustrator: the interface software for content creation.
(6) Android operating system: the platform of the AR application.

*Guide System*

By scanning the objects through this research AR guide, information will be instantly presented to people since the images of the objects have been tagged in the application. There were four kinds of interactive cultural experiences in this research AR application including: (1) history background, (2) worship rules, (3) sacred objects introduction, and (4) cultures and customs; the part of sacred objects as well as customs and cultures have an AR guide for people to enjoy.

Here are four interactive guide interfaces shown in the following:

(1) Historical background for people to get more information about Yanshui martial temple.
(2) Worship rules for people to understand worshiping and worship etiquette.
(3) Sacred objects for people to understand more about Taiwans folklore and cultural religions.
(4) Cultures and customs for people to get information about martial temple religious festivals and activities such as Yanshui fireworks.

The workflow of using the application show in Figure 1:

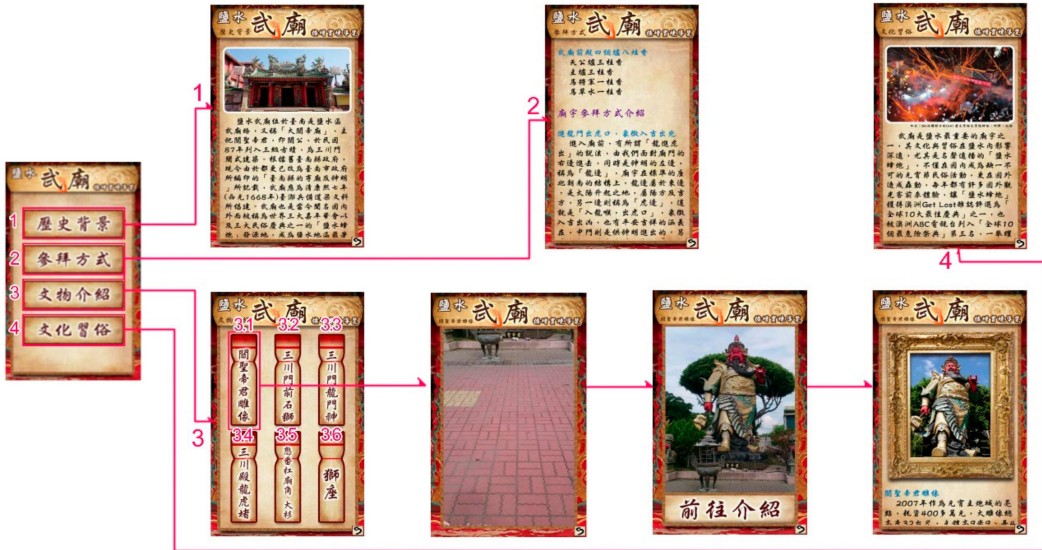

**Figure 1.** The workflow of using augmented reality (AR) interface.

Click the application interface button to get access to interactive pages. There would be buttons showing up down below the screen. After scan objects in the temple field, by clicking the buttons to get into the Guan-Gong introduction information with AR effect. Contents of the interface in the workflow of Figure 1 are described as Tables 1 and 2.

**Table 1.** The description of the interface in the AR application.

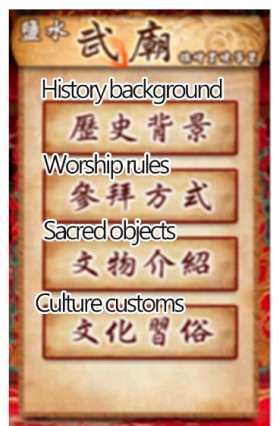

The Main menu Interface of AR guide Application for the Yansui Martial Temple

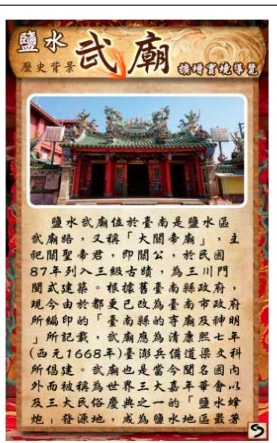

History background
Yansui martial temple is located in the Yansui area of Tainan City, also known as the Saint Guan Emperor Temple. The main God is Guan-Gong. Yansui martial temple was built in 1688. It was listed as a level three ancient monument in Taiwan, in 1988. The temple is a Fujian-style three-entry building.

**Table 1.** *Cont.*

| | |
|---|---|
| 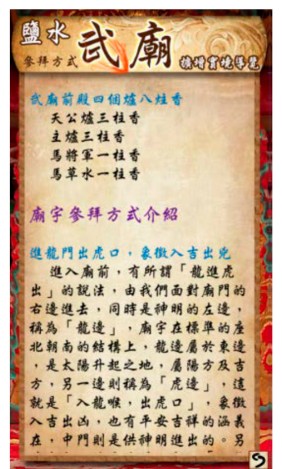 | Worship rules<br>Yansui martial temple has four incense burners in the front hall. It needs eight incense sticks total:<br>God incense burner placed three incense.<br>The main God Guan-Gong incense burner placed three incense.<br>Guan-Gong's horse, its incense burner placed an incense.<br>The horse's trough should also be placed with an incense.<br>The worship way, please follow the order,<br>That is, the door on the left is the entrance, and the door on the right is the exit. This means that you are entering the hall with the auspicious, and you can avoid disaster when you go out. |
| 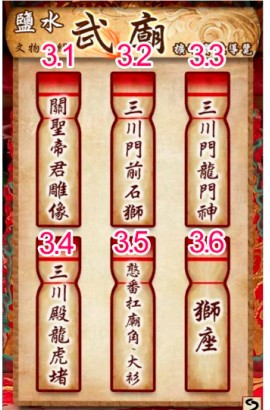 | Sacred objects<br>Interface for user to select sacred objects for information with AR, includes:<br>3.1 The Guan-Gong statue<br>3.2 The stone lion statue at front of SanChuan door<br>3.3 The door gad image of SanChuan<br>3.4 The dragon and tiger statue of SanChuan temple<br>3.5 The sculpture of the temple pillar<br>3.6 The lion statue |
| 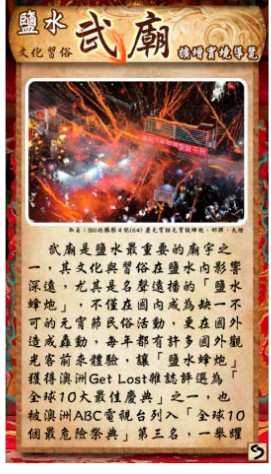 | Culture customs<br>Yanshui firework festival in Taiwan has been popular worldwide for its religious culture and customs.<br>Having been selected by the Australia's Get Lost magazine as one of the "worldwide ten best festivals", also labeled as the third out of "worldwide ten most dangerous festivals" by Australia's ABC TV. |

**Table 2.** The AR interactive images.

| | | | |
|---|---|---|---|
| 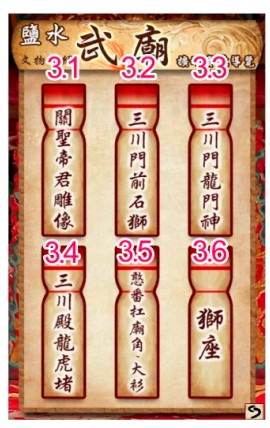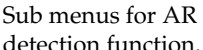 | 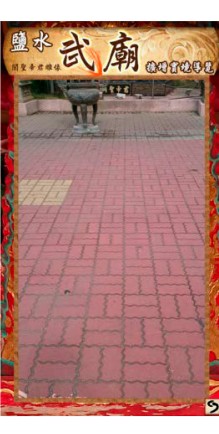 | 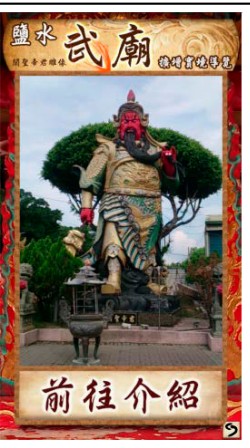 | 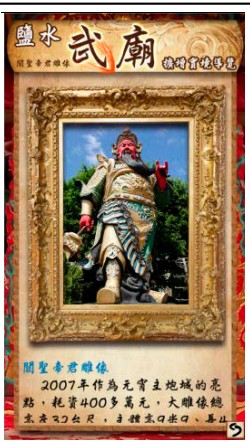 |
| Sub menus for AR detection function. | AR detection lens. | Forward to introduction. Aim the lens at the Guan-Gong statue, will response the information about the story of Guan-Gong. | The Guan gong statue. This statue was the theme of the 2007 Taiwan Lantern Festival. It cost total NT\$ 4 million to construct this lantern. The height of the statue is 32 meters. |

## 4. User Experiences Assessment

This research will examine that AR application usability and user experiences as to evaluate the users' performance in learning the cultural environment; based on theory of reasoned action (TRA) [29] and then develop the technology acceptance model (TAM) [30], and go further to take the extended technology acceptance model (TAM2) as the tools to carry out the analysis.

Compared to TAM, TAM2 has two more factors: (1) social influence process (including norms, voluntary, and images), (2) cognitive process (including career related, output quality, and performance [28]. It is clear that in TAM social norms affect user behaviors. However, through TRA, TAM, and TAM2, users' feelings and behaviors could be explained in this research.

### 4.1. Application Assessment Model

The six research factors include: (1) perceived enjoyment; (2) perceived cohesion; (3) perceived ease of use; (4) social norms; (5) user preference; (6) user loyalty. The definition of the factors is to be illustrated in the following:

(1) Perceived enjoyment

Motivation to use technology including inner and outer factors [30,31]. Inner motivation refers to the sense of satisfaction and enjoyment; as for outer motivation, will be the achievement of specific goals or rewards [31–33].

(2) Perceived cohesion

Group cohesion refers to a bond attracting people together [31,34]. For example, there will be a cohesion while people in a team have the same goals; group cohesion often relates to some behaviors such as interpersonal relationships, high degree commitments to the team, positive communication and interactions (trust), as well as regarding performance (personality similarly) [31,35–38]. Group cohesion has a significant impact on society [39,40].

(3) Perceived ease of use

According to TAM, perceived ease of use and user preference (positive attitude) is directly correlated; it should be stated that users feel relaxed in the process of experiences [39–42]. That

is to say that in case of systems that are easy to use more people are likely to perceive enjoyment. For multimedia, perceived ease of use and perceived enjoyment are highly correlated [28,39].
(4) Social norm.

Social norm refers to "the behavior standards that are set by people in consensus". The rules have been set by a majority of people in society; rules that people follow have been classified into law, moral, custom, tradition, folklore, religion, and so forth. All of which make a huge impact on society that is called social norm.

According to TAM2, the definition of social norms is "the behaviors one should do and should not do that a majority of people expect for people" [28]. In digital world the definition of social norms is dependent upon the user and their participation in groups. Based on earlier related theories, social norms will affect the part of user loyalty has been proved [29,39,43,44]. Thus, this research states that social norms are one of the factors that will affect the process of experiencing AR application.
(5) User preference

According to TRA, user preference makes a positive impact on research. The definition of Users Preference in digital technology is the acceptance of the given system [39]. Lu and Lin (2003) proposed that User preference will make a significant impact on user loyalty.
(6) User loyalty

According to Lu and Lin (2003), user loyalty is a factor that makes people keep using system. Like if the user loyalty degree is higher on a certain website one is more likely to keep using the same one [39]. This research suggests that user preference will affect user loyalty.

*4.2. Analyses and Findings*

This research has designed a questionnaire for Tainan Yanshui martial temple application users who have experienced AR guide in martial temple from 24th June, 2018 to 1st July, 2018. Seen as Figure 2. The whole process is to be executed via Google online questionnaire (https://goo.gl/forms/Ecj1bb8ZdmNvVNiv2). Out of 143 questionnaires, female users are 41.9%; male is up to 58.1%. Average age of application user is 19–29 which is about 53.4% out of all of the users. According to analyses, all of the users have a positive response to this research application. The statistical result shown as Table 3.

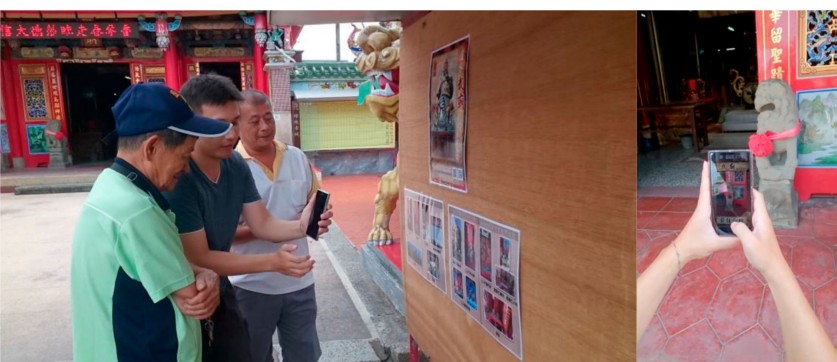

**Figure 2.** Tourists used this AR guide system.

This research used SPSS (Statistical Package for Social Science) to carry out the reliability analysis of the questionnaire. The results were: perceived enjoyment 0.904; perceived cohesion 0.928; perceived ease of use 0.875; social norm 0.790; user preference 0.844; user loyalty 0.803. Cronbach α value over 0.7 indicates a higher degree of reliability. In the analysis the lowest value among all factors is 0.79. It is clear that all factors to the research hypotheses are highly reliable [45].

**Table 3.** Statistic information.

|  | Number | Average | Standard Deviation | Variance |
|---|---|---|---|---|
| Perceived Enjoyment | 143 | 4.7395 | 0.44984 | 0.202 |
| Perceived Cohesion | 143 | 4.1923 | 0.54117 | 0.293 |
| Perceived Ease of Use | 143 | 3.9930 | 0.61877 | 0.383 |
| Social Norm | 143 | 4.4178 | 0.42870 | 0.184 |
| User Preference | 143 | 4.4462 | 0.42106 | 0.177 |
| User Loyalty | 143 | 4.2308 | 0.53695 | 0.288 |
| Effective N (complete exclusion) 143 | | | | |

### 4.3. The Analysis of the Correlation Coefficient

This research will average each factors value in the 143 questionnaires, and then takes Pearson's correlation coefficient analysis [46] to examine factor's relation to one another. When the correlation coefficient value is 1, it indicates a complete relation; 0.7–0.99 indicates both are highly correlated; 0.4–0.69 indicates that both are moderately correlated; 0.1–0.39 indicates a slight relation; 0.01–0.09 indicates both are barely correlated; when the value is 0 this indicates there is no relation. Correlation coefficient results are shown in Figure 3; each factor is significantly related to one another.

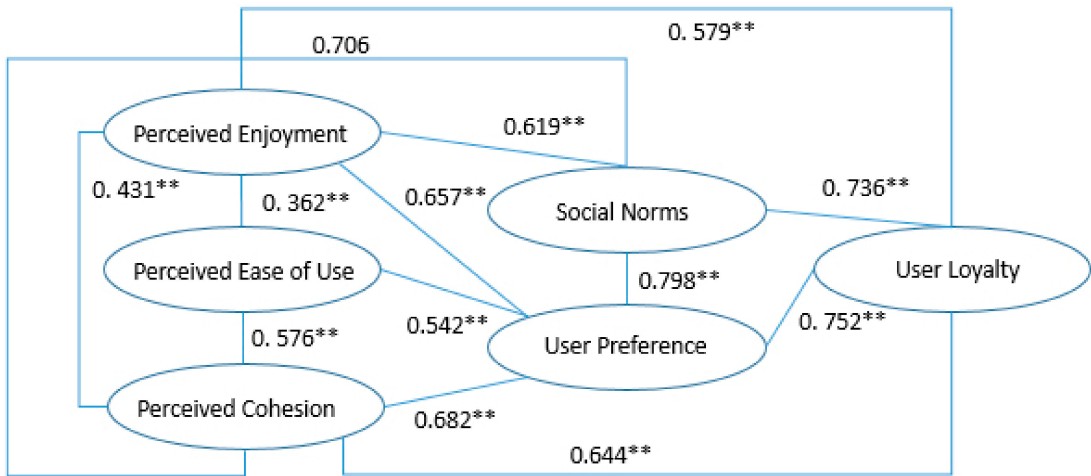

**Figure 3.** Factors relationship diagram.

Here are the analysis findings:

(1)　correlation coefficient value between perceived enjoyment and perceived cohesion was 0.431; they were moderately correlated.

(2)　correlation coefficient value between perceived enjoyment and perceived ease of use was 0.362; they were slightly correlated.

(3)　correlation coefficient value between perceived enjoyment and social norm was 0.619; they were moderately correlated.

(4)　correlation coefficient value between perceived enjoyment and user preference was 0.657; they were moderately correlated.

(5)　correlation coefficient value between perceived enjoyment and perceived user loyalty was 0.579; they were moderately correlated.

(6)　correlation coefficient value between perceived cohesion and perceived ease of use was 0.576; they were moderately correlated.

(7)   correlation coefficient value between perceived cohesion and social norm was 0.706; they were highly correlated.

(8)   correlation coefficient value between perceived cohesion and user preference was 0.682; they were moderately correlated.

(9)   correlation coefficient value between perceived cohesion and user loyalty was 0.644; they were moderately correlated.

(10)  correlation coefficient value between perceived ease of use and social norm was 0.494; they were moderately correlated.

(11)  correlation coefficient value between perceived ease of use and user preference was 0.542; they were moderately correlated.

(12)  correlation coefficient value between perceived ease of use and user loyalty was 0.446; they were moderately correlated.

(13)  correlation coefficient value between social norm and user preference was 0.798; they were highly correlated.

(14)  correlation coefficient value between social norm and user loyalty was 0.736; they were highly correlated.

(15)  correlation coefficient value between user preference and user loyalty was 0.752; they were highly correlated.

As one can see in Figure 3, user preference and social norm make a significant impact on user loyalty; as for the part of user preference, that has been influenced by social norms, perceived enjoyment, perceived ease of use, and perceived cohesion. This statistical analysis testifies that in Section 2.3 religion affects culture, shapes social life and habits, and also present people's idea and behavior. More, religious beliefs form social norms in society. Regarding using AR guide system at Tainan Yanshui martial temple, with religious belief people experience artistic and aesthetic visual pleasure and meanwhile acquire cultural knowledge. Thus, perceived cohesion (as proved in correlation analysis 7), and user preference (as proved in correlation analysis 13), and user loyalty (as proved in correlation analysis 14) should be enhanced. That is to say the part of social norms is the most important factor for the user preference and user loyalty of this AR guide research. This research AR guide system is a novel and enjoyable guide for the users who experience the sacred objects of Taiwanese culture in martial temples. According to the users, through the application, they have had an interactive experience and a better understanding of Taiwan culture and customs that they have never had before; the experience in martial temple is not only for understanding history and aesthetics but enriching spirits and education. Thus, this research states that AR application will enhance cultural experiences and learning performance.

## 5. Conclusions

The purpose of this research is to examine the users' performance in learning cultures in an interactive manner by incorporating this research AR application into martial temple; allowing people to know more about Taiwan's people's beliefs and temple culture. This research collects users' feedback (feelings and experiences) about operating AR application in martial temple from online Google questionnaire. The results of the questionnaire analysis are listed below:

(1)   according to statistical analysis, each factor average value shows that all of users have a positive response to this research AR application guide.

(2)   according to reliability analysis by SPSS software, the lowest reliable value is 0.79 that explains each factor has reached the value of reliability.

(3)   according to Pearson's correlation coefficient analysis, social norms cultural belief and user preference are highly correlated which value is up to 0.798; social norms cultural belief and user preference are significantly correlated, the value of correlation coefficient is up to 0.736. It is

obvious that social norms cultural belief is the key factor that makes impact on this research's AR application.

(4) by taking digital technology as a media, to present Taiwan temples aesthetics; images, sculptures and statues can be instantly introduced in martial temple for people to have an interactive experience and a better understanding of Taiwan cultures.

**Author Contributions:** Conceptualization, Y.-S.C.; formal analysis, Y.-S.C.; methodology, Y.-S.C.; writing—original draft, Y.-S.C. and Y.-J.R.H.; writing—review & editing, Y.-S.C. and Y.-J.R.H.; supervision, Y.-S.C.; software, validation, H.-W.C.; visualization H.-W.C.

**Funding:** This research received no external funding.

**Conflicts of Interest:** The authors declare no conflict of interest.

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
