# Peer review of "Learning Performance Assessment for Culture Environment Learning and Custom Experience with an AR Navigation System"

_sustainability, doi:10.3390/su11174759_

Round 1

Reviewer 1 Report

1. In the aspect of originality, academia, and applicable value: this paper (literature) takes Taiwan cultural environment education as the theme to present Tainan temples cultures, connotations, and religions; with AR interactive devices , reach the goal of sustainable cultural environment. This research takes APP users as the research objects; carry out the questionnaire survey as to obtain the feedback of the users cultural experiences. By means of empirical analysis to testify the Usability of the system.   The contribution of this research is based on a brand-new type AR interactive technology (innovative )development for the field  of research on the education of culture of cultural environment education . on real field through AR interactive technology    2. Content: regarding Taiwan Cultures and Religions in 2-1, suggest add more related literature reviews. And, in the part of AR APP evaluation, suggest make illustration of how Perceived Cohesion and Social Norms affect the cultural religions. 3. System usability should be stated in literature reviews; what is the standard for evaluation ? 4. Research method and hypothesis ( strict level), writer applies TAM model as the research tool to meet the digital technology features, and takes the part of social norms and perceived cohesion that are related to cultural religions  to examine the system usability six factors: (1) Perceived Enjoyment; (2) Perceived Cohesion; (3) Perceived Ease of Use; (4) Social Norms; (5) User Preference; (6) User Loyalty. This research takes Pearson’s correlation coefficient analysis to testify if the six factors above  exist or related to one another. By which to examine the usability of AR APP for users to experience cultural environment learning . Research method is quantitative statistic analysis, the results are precise. In 4.3, the writer states each factor if exist and if have positive relation with one another, yet without the details of the relationships between each factor , thus suggest the writer should make a illustration for the factors relations as well as the sustainable cultural environment learning.

Author Response

Point 1: 1. In the aspect of originality, academia, and applicable value: this paper (literature) takes Taiwan cultural environment education as the theme to present Tainan temples cultures, connotations, and religions; with AR interactive devices , reach the goal of sustainable cultural environment. This research takes APP users as the research objects; carry out the questionnaire survey as to obtain the feedback of the users cultural experiences. By means of empirical analysis to testify the Usability of the system, the contribution of this research is based on a brand-new type AR interactive technology (innovative )development for the field  of research on the education of culture of cultural environment education . on real field through AR interactive technology.

Response 1: Thanks to the Reviewer 1 for the contribution in the originality, academic and practical value of this submitted paper.

Point 2: Content: regarding Taiwan Cultures and Religions in 2-1, suggest add more related literature reviews. And, in the part of AR APP evaluation, suggest make illustration of how Perceived Cohesion and Social Norms affect the cultural religions.

Response 2: In paragraph 2 of Section 2.1, we added related literature reviews as follows: Culture and religion always exist in a close relation. Together with aesthetics and ethics, religion constitutes culture. Moreover, religion as cultural identity marker, it caused the borders between culture and religion to blur. To understand culture can be about social culture and religion [6]. Mulder (1985) indicates that studying religion implies religion as an expression of human culture. Religion is, thus, expressed and clothed in cultural guise. Comprehending religion then implies studying human culture [7].

Point 3: System usability should be stated in literature reviews; what is the standard for evaluation?

Response 3: In the literature review we added the 2.3 section for the descriptions of The Usability Assessments; and the theoretical basis for the six evaluated constructs.

Point 4: Research method and hypothesis (strict level), writer applies TAM model as the research tool to meet the digital technology features, and takes the part of social norms and perceived cohesion that are related to cultural religions  to examine the system usability six factors: (1) Perceived Enjoyment; (2) Perceived Cohesion; (3) Perceived Ease of Use; (4) Social Norms; (5) User Preference; (6) User Loyalty. This research takes Pearson’s correlation coefficient analysis to testify if the six factors above  exist or related to one another. By which to examine the usability of AR APP for users to experience cultural environment learning . Research method is quantitative statistic analysis, the results are precise. In 4.3, the writer states each factor if exist and if have positive relation with one another, yet without the details of the relationships between each factor , thus suggest the writer should make a illustration for the factors relations as well as the sustainable cultural environment learning.

Response 4:In 4.3, the authors added a high correlation causal description about the relationship between Perceived Cohesion, Social Norms for User Preference and User Loyalty of using the AR guide system. It is also pointed out that Social Norm is the most important factor in the success of cultural environment learning using the AR guide system.

Reviewer 2 Report

The paper is interesting and deals with a relevant subject. However, the English requires extensive rewrite in order to be applicable for publication.

Issues:

There is a large emphasis in the introductory chapters on explaining religious aspects like the Gods and customs. This can be reduced, keeping only enough to explain the problem that this research is tackling. The word "APP" should be written as "application". Chapter 2.1. should have a few more references (for example lines 88 - 89), same for lines 95 and 96 in the next chapter. Chapter 3.1. should have more details describing the application created. Try to explain the workflow of using the application in more detail. Line 168 has an opening bracket which is never closed. Line 216 needs a scientific reference and explanation for SPSS and the full name, not just the abbreviation. Line 222 Pearson's correlation needs a scientific reference.

Author Response

Response to Reviewer 2 Comments

Point 1: There is a large emphasis in the introductory chapters on explaining religious aspects like the Gods and customs. This can be reduced, keeping only enough to explain the problem that this research is tackling.

Response 1: The authors had corrected the contents of Chapter1 and reached a clear thesis statement

Point 2: The word "APP" should be written as "application".

Response 2: The paper has been revised according to the reviewer's opinion

Point 3: Chapter 2.1. should have a few more references (for example lines 88 - 89), same for lines 95 and 96 in the next chapter.

Response 3: Lines 88 – 89, lines 95 and 96 citations have been added. And, in paragraph 2 of Chapter 2.1, we added related literature reviews as follows: Culture and religion always exist in a close relation. Together with aesthetics and ethics, religion constitutes culture. Moreover, religion as cultural identity marker, it caused the borders between culture and religion to blur. To understand culture can be about social culture and religion [6]. Mulder (1985) indicates that studying religion implies religion as an expression of human culture. Religion is, thus, expressed and clothed in cultural guise. Comprehending religion then implies studying human culture [7].

Point 4: Chapter 3.1. should have more details describing the application created. Try to explain the workflow of using the application in more detail.

Response 4: Description for system development tools and creation have been added in Chapter 3. The workflow of using the application is as follows (seen as figure 1).

Point 5: Line 168 has an opening bracket which is never closed. Line 216 needs a scientific reference and explanation for SPSS and the full name, not just the abbreviation. Line 222 Pearson's correlation needs a scientific reference.

Response 5: These above errors have been fixed.

Reviewer 3 Report

Originality/Novelty: The main theme of the article is the application of AR in the enhancement of the cultural tourism experience.  The results cannot really be considered breakthrough advances as the capability of AR or ICT in providing heightened experiences is well-known.

Significance: The authors provide a clear thesis statement, define the key concepts, and substantiate their hypothesis by a sound statistical analysis.

Quality of Presentation: The article contains several stylistic and composition-related errors significantly interfering with its readability and interpretability.

Scientific Soundness: The study is logically and proportionally structured and the authors provide detailed description of the instruments and software deployed. The article follows the guidelines and rules of scholarly publication.

 Interest to the Readers: Regarding the original context of the journal focusing on educational assessment and technology, the article is tangentially related at best. While the application of AR for educational assessment is not treated, the article places a heavy emphasis on technology. The authors describe the background, actual process, and results of an intriguing experiment.

Overall Merit: The paper deals with an important development of ICT technology, namely the application of AR in cultural tourism. The greatest value of the text is the thorough statistical analysis and the depth to which the given results are interpreted.

English Level: The frequent grammatical and stylistic errors make reading and processing difficult.

Author Response

Response to Reviewer 3 Comments

Point 1: Originality/Novelty: The main theme of the article is the application of AR in the enhancement of the cultural tourism experience.  The results cannot really be considered breakthrough advances as the capability of AR or ICT in providing heightened experiences is well-known.

Response 1: Thanks to the reviewer's suggestion, this paper verifies that the AR guide application provides a digital tool for cultural environment learning, and users can affirm the usability of AR applications because of their religious beliefs and social customs.

Point 2: Significance: The authors provide a clear thesis statement, define the key concepts, and substantiate their hypothesis by a sound statistical analysis.

Response 2: Thanks to the reviewer for your affirmation of the article.

Point 3: Quality of Presentation: The article contains several stylistic and composition-related errors significantly interfering with its readability and interpretability.

Response 3: Thanks to the reviewer for the statement correction for this article. The author has made corrections based on sustainability-573658-review.pdf.

Point 4: Scientific Soundness: The study is logically and proportionally structured and the authors provide detailed description of the instruments and software deployed. The article follows the guidelines and rules of scholarly publication.

Response 4: Thanks to the reviewer for your affirmation of the article.

Point 5: Interest to the Readers: Regarding the original context of the journal focusing on educational assessment and technology, the article is tangentially related at best. While the application of AR for educational assessment is not treated, the article places a heavy emphasis on technology. The authors describe the background, actual process, and results of an intriguing experiment.

Response 5: Thanks to the reviewer for your affirmation of the article.

Point 6: Overall Merit: The paper deals with an important development of ICT technology, namely the application of AR in cultural tourism. The greatest value of the text is the thorough statistical analysis and the depth to which the given results are interpreted.

Response 6: Thanks to the reviewer for your affirmation of the article.

 Point 7: English Level: The frequent grammatical and stylistic errors make reading and processing difficult.

Response 7: The authors have made modifications based on the review comments provided by reviewer. This article will still perform proofreading.
